# IC-Mapper: Instance-Centric Spatio-Temporal Modeling for Online Vectorized Map Construction

Submission Id: *****

## ABSTRACT

Online vector map construction based on visual data can bypass the processes of data collection, post-processing, and manual annotation required by traditional map construction, which significantly enhances map-building efficiency. However, existing work treats the online mapping task as a local range perception task, overlooking the spatial scalability required for map construction. We propose *IC-Mapper*, an instance-centric online mapping framework, which comprises two primary components: 1) **Instance-centric temporal association module:** For the detection queries of adjacent frames, we measure them in both feature and geometric dimensions to obtain the matching correspondence between instances across frames. 2) **Instance-centric spatial fusion module:** We perform point sampling on the historical global map from a spatial dimension and integrate it with the detection results of instances corresponding to the current frame to achieve real-time expansion and update of the map. Based on the nuScenes dataset, we evaluate our approach on detection, tracking, and global mapping metrics. Experimental results demonstrate the superiority of IC-Mapper against other state-of-the-art methods.

## CCS CONCEPTS

• **Do Not Use This Code → Generate the Correct Terms for Your Paper**; *Generate the Correct Terms for Your Paper*; Generate the Correct Terms for Your Paper; Generate the Correct Terms for Your Paper.

## KEYWORDS

End-to-end online map construction; Temporal association; Spatial fusion; Detection and tracking

**ACM Reference Format:**
Anonymous Author(s). 2024. IC-Mapper: Instance-Centric Spatio-Temporal Modeling for Online Vectorized Map Construction. In *Proceedings of Make sure to enter the correct conference title from your rights confirmation emai (Conference acronym 'XX).* ACM, New York, NY, USA, 9 pages. https://doi.org/XXXXXXX.XXXXXXX

## 1 INTRODUCTION

High-definition (HD) maps contain rich static vector elements of traffic scenes such as lane lines, boundaries, and crosswalks,

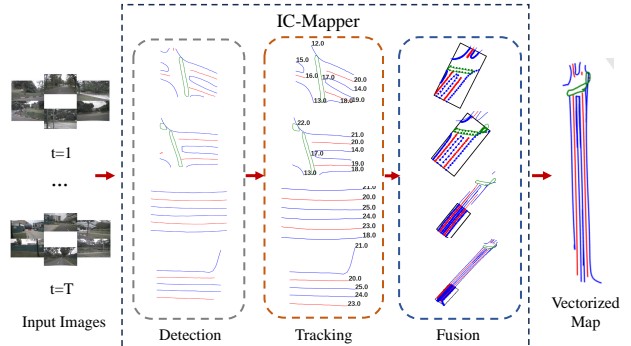

**Figure 1: Traditional deep learning-based online vector map construction approaches focus only on local detection performance. By incorporating temporal tracking and spatial fusion modules, we have implemented an end-to-end detection-tracking-fusion process that enables the construction of global maps.**

which provides information beyond the perception range for autonomous driving vehicles, and is crucial for tasks such as localization, decision-making and trajectory planning. However, traditional processes for generating and maintaining HD maps are exceedingly complex, involving the pre-collection of a large amount of point cloud and image data, followed by offline feature extraction, post-processing optimization, manual annotation, and quality inspection. While this can achieve high-accuracy maps, such a complex process makes long-term maintenance and updates not only costly but also difficult to quickly adapt to new environments, which limits the development of autonomous driving technology in unknown settings. Therefore, there is a growing urgency for affordable vision-based online map reconstruction technique.

Most works treat the task of online vectorized map reconstruction as a local range perception task Early efforts firstly detect static features such as lane lines in images, then use camera parameters and Inverse Perspective Mapping (IPM) to project detection results into the vehicle's coordinate system [25]. In recent years, advancements in perception technologies have led to studies utilizing existing algorithms based on Bird's-Eye-View (BEV) representations to directly generate BEV space detection results from image data in an end-to-end fashion. [12, 16]. HDMapNet [14] is the first attempt to use images from multiple cameras as the input on the nuScenes [5] dataset, employing a neural network to provide local BEV space map segmentation results on an end-to-end basis. Subsequently, it utilizes post-processing methods such as clustering and fitting to transform the segmentation maps into vectorized representations. VectorMapNet [18] and MapTR [17], by designing detection decoder structures similar to DETR [7], achieve the end-to-end output of vectorized map representation. Further, StreamMapNet [34] introduces temporal modeling by using multi-frame images as the input

to enhance the accuracy of map instance detection. However, existing deep learning-based approaches restrict the task of detecting map instances within a fixed BEV range, failing to achieve real-time construction and updates of a global map.

Addressing this gap in research, as shown in Fig. 1, this paper introduces an end-to-end online vector map construction method that takes a temporally continuous sequence of images as the input and outputs global vectorized map construction results in an end-to-end manner. Nevertheless, constructing spatially continuous maps from temporally continuous observational data using deep learning is a challenging task. The primary difficulties are manifested in two aspects: (1) **Capturing the temporal association of map instances.** Identifying the same map instances across frames is crucial for subsequent fusion and merging to construct a global map. However, in complex traffic environments, the same map instance often appears differently in different frames due to the ego vehicle's movement, obstructions, changes in lighting, and other factors. (2) **Maintaining the spatial consistency of the map.** Map instances such as lane dividers and road boundaries often span large spatial scales, whereas the observational data from each frame can only cover a local range with the conflict between different frames.

To address the difficulties outlined above, we propose IC-Mapper, an end-to-end vector map detection, tracking, and fusion scheme for online map updating and construction. Building on an existing vector map detection network, we first introduce an **instance-centric temporal association module** that matches detected instances with tracked instances in both geometric and feature dimensions, obtaining correspondence between map instances in adjacent frames. Secondly, we introduce an **instance-centric spatial fusion module** that performs sampling and encoding on the map maintained from previous frames in the spatial dimension and then executes fusion operation with the current detection results. Ultimately, the fused results are updated to the map, achieving continuous map construction. Our main **contributions** are as follows:

- We propose an end-to-end framework that for the first incorporates all online mapping tasks in one network including the detection, tracking and the global map update.
- We introduce an instance-centric temporal association module and a spatial fusion module to enable online tracking and fusion of map instances.
- We extensively evaluate the online vectorized mapping tasks. Experimental results further illustrate the state-of-the-art performance of our method compared to other mapping approaches across diverse metrics.

## 2 RELATED WORK

### 2.1 Visual Multiple Object Tracking

Traditional multi-object tracking algorithms perform the object associations based on detection results. Some early formulations [2, 3, 27] rely on Kalman filters [31] or optical flow algorithms [1] to estimate and match moving targets' positions from one frame to the next. Another kind of approach [29, 37] involves training additional target re-identification networks to assign the same ID to targets with similar features. In recent years, with the development of the Transformer [28], query-centered end-to-end detection and tracking methods [35, 36] have been proposed, eliminating

the complex post-processing and matching computations found in traditional methods. In the field of HD mapping, tracking modules are typically independent. However, experimental evidence suggests that performing tracking tasks end-to-end can compromise detection performance. Hence, recent vectorized HD map construction works, like MOTRv3 [33] and DQTrack [15], take significant efforts to address this issue, while there is little work on using deep learning-based end-to-end tracking algorithms in mapping tasks.

### 2.2 Vectorized HD Map Construction

Offline map construction requires pre-collection of sensor data for the corresponding environment, followed by offline execution of three steps: map feature extraction, vectorized feature modeling, and map updating. Map feature extraction aims to extract map-related target features from image data, utilizing traditional geometric detection algorithms [22, 23, 32], as well as deep learning-based semantic segmentation [13, 38] or object detection algorithms [11, 30]. The goal of vectorized feature 3D modeling is to transform features extracted from images into 3D space, traditionally achieved through IPM projection or using synchronized point cloud data. In recent years, deep learning-based 3D reconstruction algorithms [19] have also been used for feature modeling. Map updating aims to fuse features extracted from multiple frames. Some methods achieve this using traditional rule-based algorithms [9, 26], while recent approaches also fuse multiple frame outputs using deep learning-based methods [39]. Overall, offline methods offer high-accuracy map reconstruction but are cumbersome and prone to error accumulation from various modules.

*2.2.1 Vision-Based Online Mapping.* Vision-based online vector map construction is gaining attention for its real-time updates in dynamic environments and mapping efficiency. Traditional vision SLAM (Simultaneous Localization and Mapping) systems [6, 20, 21] use generic feature points for front-end representation, making it challenging to directly create vectorized maps. Qiao and others [24] construct a tracking and optimization module based on lane line representation for online lane map updates by using 3D lane line results obtained from a pre-trained detector [8]. In recent years, deep learning-based end-to-end map construction tasks have also garnered widespread interest. HDMapNet [14], based on surround-view images, achieves end-to-end map instance segmentation tasks, followed by post-processing to complete vectorized instances extraction. VectorMapNet [18] and MapTR [17] accomplish end-to-end network output of vectorized map representations by designing decoders based on DETR [7]. StreamMapNet [34] further integrates temporal multi-frame information to enhance the accuracy of detection. However, existing deep learning-based online end-to-end map construction methods only focus on detecting map instances within a local perception range, unable to perform real-time updates and construction of global maps.

## 3 METHOD

### 3.1 Overview

This section provides an overview of the IC-Mapper. The input of the model is the synchronized multi-view image sequences denoted by $I = \{I_t\}_{t=1}^{T}$. In this context, $I_t$ denotes the set of images

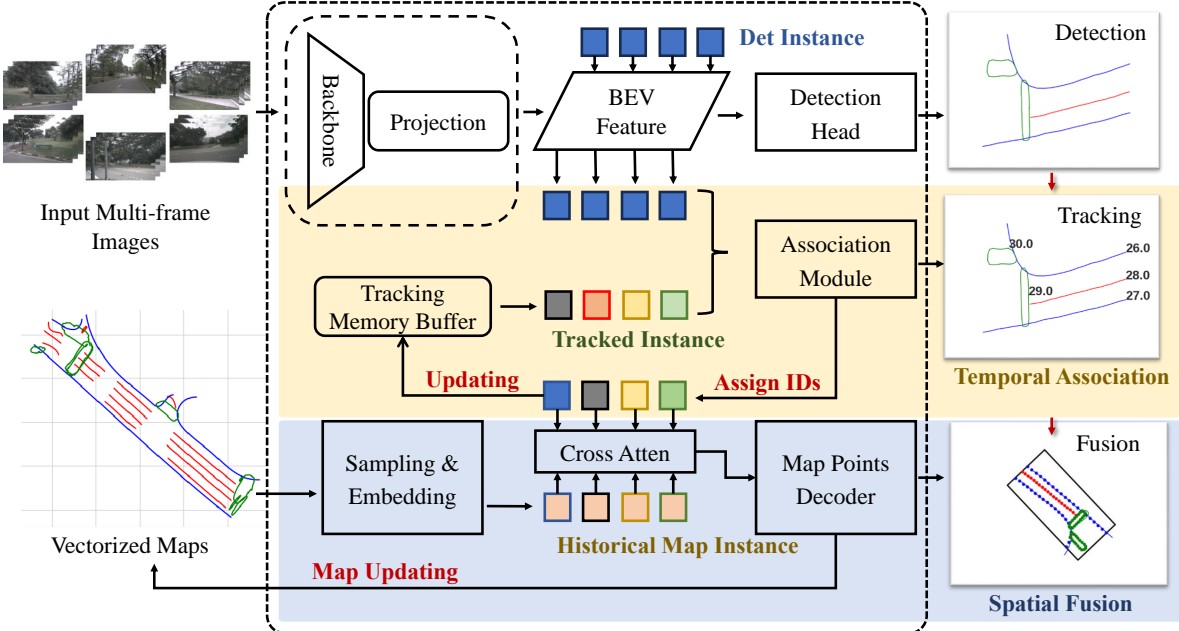

**Figure 2: The overall framework of IC-Mapper. The input consists of continuous multi-frame surround-view images. Building upon an existing query-based visual detector, we introduce an instance-centric temporal association module and a spatial fusion module, which enables end-to-end detection, tracking, and fusion of map instances and in turn facilitates the online reconstruction of global vectorized maps.**

obtained at time step $t$, and $T$ indicates the total count of these time steps. The output comprises a series of vectorized map instances $\mathcal{S}_G^{1:T} = \{(c_i, P_i, id_i)\}_{i=1}^{N_G^{1:T}}$, corresponding to all the regions covered by the vehicle. Here, $N_G^{1:T}$ signifies the total count of map instances within the scanned area, where $c_i$ identifies the category of the $i$-th instance, and $P_i \in \mathbb{R}^{N_p \times 2}$ denotes the associated point set where $N_p$ is the number of points. $id_i$ provides a unique identifier for each instance. The entire architecture, as shown in Fig. 2, builds upon the detector designed concerning StreamMapNet [34] and incorporates instance-based temporal association and spatial fusion modules to achieve end-to-end map detection, tracking, and updating.

## 3.2 Basic Detection Module

The detection module, building on StreamMapNet [34], generates BEV (Bird's Eye View) feature maps concerning BEVFormer [16]. A DETR-based [7] decoder is designed to refine the points of map instances iteratively through a multi-layer Transformer. Furthermore, for temporal modeling, a GRU-based [10] approach is used for fusing BEV features, along with a query feature fusion method grounded in the TopK mechanism. To transition the instance's query features from one frame to the next, a Multi-Layer Perception (MLP) based query update module, $U_{MLP}$, is employed, which is supervised by the L1 distance between decoded coordinate points. The corresponding loss function is denoted as $\mathcal{L}_{trans}$. Given the detected map instances $\mathcal{D}^t = \{(c_i, P_i)\}_{i=1}^{N_q^t}$ and the ground truth set $\mathcal{G}^t = \{(\hat{c}_i, \hat{P}_i)\}_{i=1}^{N_{gt}^t}$, where $N_q$ and $N_{gt}$ are the number of detection queries and ground truth, we firstly employ bipartite matching to get the optimal instance-level and point-level assignment ($\hat{\pi}$ and

$\hat{\gamma}_i$) as in [17]. Then the detection loss is defined as:

$$\mathcal{L}_{det} = \lambda_1 \sum_{i=0}^{N_{gt}-1} \mathcal{L}_{\text{Focal}}(c_{\hat{\pi}(i)}, \hat{c}_i) + \tag{1}$$

$$\lambda_2 \sum_{i=0}^{N_{gt}-1} \mathbb{1}_{\{c_i \neq \varnothing\}} \sum_{j=0}^{N_p-1} D_{\text{SmoothL1}}(P_{\hat{\pi}(i),j}, \hat{P}_{i,\hat{\gamma}_i(j)}), \tag{2}$$

where $\hat{\pi}_i$ denotes the index of the detected instance corresponding to the $i$-th ground truth (GT) instance under optimal matching. Conversely, $\hat{\gamma}_i(j)$ indicates the index of the $j$-th point in the $\hat{\pi}_i$-th detected instance that matches the coordinates of point $\hat{P}_i$ under optimal matching.

## 3.3 Instance-Centric Temporal Association

In the temporal association module, as shown in Fig. 3, for the current frame $t$, we maintain a tracking memory buffer $\mathcal{A}_{1:t-1} = \{(c_i, P_i, Q_i, id_i)\}_{i=1}^{N_{\mathcal{A}}^{1:t-1}}$ to store $N_{\mathcal{A}}^{1:t-1}$ instances tracked in previous frames. The set of detected instances in the current frame $t$ is $\mathcal{R}_{det}^t = \{(c_i, P_i, Q_i)\}_{i=1}^{N_{det}^t}$, where $Q_i$ represents the query feature of the $i$-th instance. The temporal association module learns the matching relationships between instances in $\mathcal{A}^{1:t-1}$ and $\mathcal{R}_{det}^t$, subsequently assigning ID information to the instances in $\mathcal{R}_{det}^t$ and updating it to $\tilde{\mathcal{R}}_{det}^t = \{(c_i, P_i, Q_i, id_i)\}_{i=1}^{N_{det}^t}$.

*3.3.1 Learnable Temporal Map Instance Association.* To establish comprehensive matching relationships between instances, we model the relationships of detected and tracked instances through both

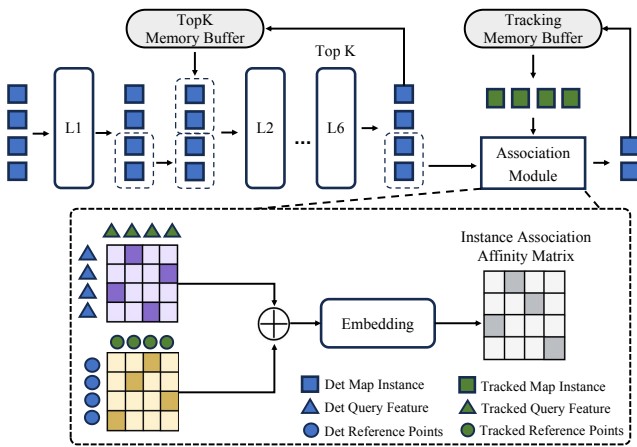

**Figure 3: The temporal association module. The top $k$ refined queries are propagated from the previous frame. Following their transformation, these queries are combined with $N_q^t - k$ queries from the current frame, resulting in an updated ensemble of $N_q^t$ queries. Subsequently, this updated query set is dynamically assimilated into the tracking memory buffer via the instance corresponding module.**

geometric and feature dimensions. Specifically, we use $Q_D \in \mathbb{R}^{N \times D}$ as the query feature embeddings for the detected instances in the current frame. And $Q_T \in \mathbb{R}^{M \times D}$ represents the query feature of tracked instances from Tracking Memory Buffer, which have been updated by $U_{MLP}$. Here, $N$ and $M$ represent the number of detected and tracked instances, respectively, while $D$ denotes the feature dimension. Similarly, $\mathcal{P}_D \in \mathbb{R}^{N \times N_p \times 2}$ and $\mathcal{P}_T \in \mathbb{R}^{M \times N_p \times 2}$ are used to represent the point sets corresponding to the detected and tracked instances in the current frame's ego-vehicle coordinate system, respectively. In the geometric dimension, we encode the L2 metric matrix between $\mathcal{P}_D$ and $\mathcal{P}_T$ using a Feed-Forward Network (FFN); in the feature dimension, we encode the outer product of $Q_D$ and $Q_T$ using a Multi-Layer Perceptron (MLP). The encoding results from both dimensions are summed and then passed through another layer of MLP to obtain the fused metric relationship encoding result $\mathcal{H} \in \mathbb{R}^{N \times M \times 1}$.

$$\mathcal{H} = \text{MLP}(\text{MLP}(Q_D \odot Q_T) + \text{FFN}(\text{L2}(\mathcal{P}_D, \mathcal{P}_T))), \quad (3)$$

where each position in $\mathcal{H}$ is used to measure the similarity between the corresponding detected and tracked instances.

*3.3.2 Optimization and Inference.* In this section, we focus on more details about model optimization and inference on map instance tracking with the help of above designed learnable association. **Optimization Objectives.** Based on the ground truth (GT) ID labels, we construct a real binary metric matrix $\mathcal{G} \in \mathbb{R}^{N \times M}$ to represent the matching relationships between detected and tracked instances. Each element of $\mathcal{G}$ is set to 1 if the corresponding detected instance (row index) and tracked instance (column index) match according to the GT, and 0 otherwise. Then, the cross-entropy loss between $\mathcal{H}$ and $\mathcal{G}$ is calculated to serve as the loss for the temporal

association module:

$$\mathcal{L}_{\text{asso}} = -\sum_{i=1}^{N}\sum_{j=1}^{M}\left[\mathcal{G}_{ij}\log(\mathcal{H}_{ij}) + (1 - \mathcal{G}_{ij})\log(1 - \mathcal{H}_{ij})\right]. \quad (4)$$

Here, $\mathcal{G}_{ij}$ represents the ground truth matching between the $i$-th detected instance and the $j$-th tracked instance. This approach effectively trains the temporal association module to improve its accuracy in matching instances over time. **Inference Scheme.** During the inference process, upon obtaining the similarity score matrix $\mathcal{H} \in \mathbb{R}^{N \times M \times 1}$, which delineates the affinity between current detection instances and historical tracking instances, the ID assignment to current detection instances can be formalized through the following steps:

1. Threshold Application: A predefined threshold, $\theta$, is applied to the scores in $\mathcal{H}$, such that only those detection-tracking instance pairs with scores exceeding $\theta$ are considered for further processing.

$$\mathcal{H}_{filtered} = \{\mathcal{H}_{ij} | \mathcal{H}_{ij} > \theta, \forall i, j\}. \quad (5)$$

2. Optimal Matching: The most compatible detection-tracking pairs are selected based on the maximization of the total similarity score across all matches. This step ensures a one-to-one correspondence between detections and tracks, where each detection $i$ is matched with at most one track $j$ and vice versa, formalized as:

$$\mathcal{M} = \text{argmax}_{\mathcal{H}_{filtered}} \sum_{i,j} \mathcal{H}_{ij}, \quad (6)$$

where $\mathcal{M}$ denotes the set of matched detection-tracking instance pairs.

3. ID Allocation: Each detection instance $i$ in the current frame inherits the ID of its matched tracking instance $j$ from $\mathcal{M}$. Detection instances without a match are assigned new IDs, indicating the emergence of new objects.

$$ID(i) = \begin{cases} ID(j) & \text{if } (i, j) \in \mathcal{M}, \\ \text{new ID} & \text{otherwise.} \end{cases} \quad (7)$$

4. Tracking Buffer Update: The tracking memory buffer is updated by $\tilde{\mathcal{R}}_{det}^t$ to reflect the current frame's detection instances and their assigned IDs, incorporating the additions of new objects and removal of unmatched tracks.

## 3.4 Instance-Centric Spatial Fusion and Updating

To construct a global map that contains spatially continuous map instances, we further employ a spatial fusion module that fuses map instances from the current frame into the maintained global map. In this module, we firstly sample a fixed number of points from the intersection of the historical global map $\mathcal{S}_G^{1:t-1} = \{(c_i, P_i, id_i)\}_{i=1}^{N_G^{1:t-1}}$ with the current perception range. These sampled points are then fused one-to-one with instances in $\tilde{\mathcal{R}}_{det}^t$ through cross-attention operations. Furthermore, we introduce a class-wise merging strategy to smoothly insert the upcoming detection result into the final global map $\mathcal{S}_G^{1:t}$. For the sake of brevity, we remove the id symbol $id$ from map instances $\mathcal{S}_G = \{(c_i, P_i, id_i)\}_{i=1}^{N^t}$ since the following fusion and merge are performed based on tracking results. An illustration of this process is shown in Fig. 4.

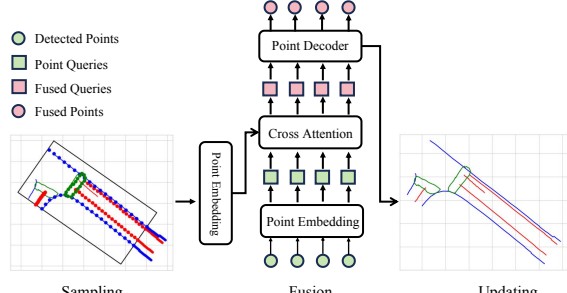

**Figure 4: An illustration of the instance-centric spatial fusion and updating process. Before fusion, we first sample the history point sets around the intersection area between the current patch and the maintained global map. Then a cross-attention-based spatial fusion is applied between the detected points and the sampled historical points. The fused queries are further decoded to serve as the final result, which is then updated into the global map using a curve-fitting-based merging algorithm.**

*3.4.1 Spatial Point Set Sampling.* During training and inference, a global map $\mathcal{S}_G$ is maintained for an entire scene. It comprises point sets whose coordinates are in the world frame. Before sampling, we first define a rectangular patch $\text{Patch}_{\text{cur}}$ centered on the ego pose of the current frame. Its range is firstly expanded from the model's perception range by 20m in BEV space to obtain more overlapped areas with historical frames. For each global map instance that contains historical point sets, we first calculate the intersection sets between the current patch and the historical point sets, denoted as $\mathcal{P}^{\text{inter}}$. The corresponding set operation is denoted as $\text{Intersection}(\text{Patch}, \mathcal{P})$ and $\text{Diff}(\text{Patch}, \mathcal{P})$, respectively. Finally, we perform an evenly spaced sampling on the intersection point sets. We denote the evenly spaced sampling operation as $\text{Sampling}(\mathcal{P})$. This point sampling algorithm is shown in Algorithm 1.

---

**Algorithm 1** Point Sampling Algorithm

---

**Input:** $\text{Patch}_{\text{cur}}, \mathcal{S}_G = \{(c_i, \mathcal{P}_i)\}_i^{N_G}$

**Output:** Sampled historical points $\{\mathcal{P}_i^{\text{sample}}\}_{i=1}^{N^t}$ corresponding to each detected map instance
1: **for** $\mathcal{S}_j(c_j, \mathcal{P}_j) \in \mathcal{S}_G$ **do**
2:  Expand current patch to $\text{Patch}_{\text{cur}}^{expand}$.
3:  $\mathcal{P}_j^{\text{inter}} = \text{Intersection}(\text{Patch}_{\text{cur}}^{expand}, \mathcal{P}_j)$.
4:  $\mathcal{P}_j^{\text{sample}} = \text{Sampling}(\mathcal{P}_j^{\text{inter}})$.
5: **end for**
6: **return** $\{\mathcal{P}_i^{\text{sample}}\}_{i=1}^{N^t}$

---

*3.4.2 Map Instance Spatial Fusion.* After retrieving the corresponding sampled historical points $\{\mathcal{P}_i^{\text{sample}}\}_{i=1}^{N^t}$ for each detected map instance of the current frame, we employ 3 transformer layers where cross attention is calculated between the detected point sets $\{\mathcal{P}_i\}_{i=1}^{N^t}$ and the sampled historical points $\{\mathcal{P}_i^{\text{sample}}\}_{i=1}^{N^t}$. For the sake of clarity and brevity, we reformat these point sets as tensors

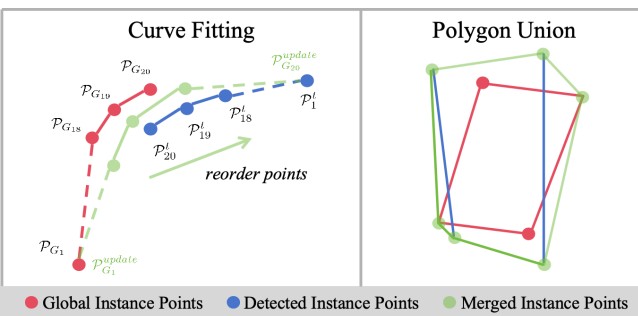

**Figure 5: An illustration of the curve fitting algorithm. As can be seen, polyline-type instances are merged based on curve fitting while polygon-type instances are merged simply by union. During curve fitting, points are first reordered and concatenated. Then a fitting and resampling is performed to generate the updated point sets. We denote the points from the global map as** red**, points from detected instances as** blue**, and the final points as** green**.**

omitting the instance index, denoted as $P_k \in \mathbb{R}^2$ where $k$ represents the $k$-th layer, and we omit the self-attention and feed-forward network components. Point tensors are firstly encoded by several MLP layers and then standard multi-head cross attention (denoted as $\text{CrossAttn}(Q, K, V)$, where $Q$, $K$ and $V$ represents query, key, and value ) is applied:

$$Q_{k-1} = \text{Point\_Embed}(P_{k-1}), \tag{8}$$

$$V_{k-1} = \text{Point\_Embed}(P_{k-1}^{\text{sample}}), \tag{9}$$

$$K_{k-1} = \text{Point\_Embed}(P_{k-1}^{\text{sample}}), \tag{10}$$

$$Q_k = \text{CrossAttn}(Q_{k-1}, K_{k-1}, V_{k-1}), \tag{11}$$

$$P_{k+1} = \text{Reg}(Q_k). \tag{12}$$

To be noticed, following [34], a shared regression layer Reg is employed for all layers to predict the final point coordinates. The predicted points from the current layer serve as the query points input to the next layer. The output of the last layer is the final refined detection result which is used to infer an additional fusion loss. Here we simply employ a regression loss:

$$\mathcal{L}_{fusion} = \sum_{i=0}^{N^t-1} \mathcal{L}_{\text{SmoothL1}}(P_i, \hat{P}_i), \tag{13}$$

where $\hat{P}_i$ is the ground truth point set corresponding to the predicted instance.

Hence, the final loss is the sum of the detection loss $\mathcal{L}_{det}$, transition loss $\mathcal{L}_{trans}$, the tracking loss $\mathcal{L}_{track}$ and the fusion loss $\mathcal{L}_{fusion}$:

$$\mathcal{L} = \mathcal{L}_{det} + \alpha \mathcal{L}_{trans} + \beta \mathcal{L}_{asso} + \lambda \mathcal{L}_{fusion}. \tag{14}$$

This loss serves as the final end-to-end training target, where $\alpha$, $\beta$, and $\lambda$ are the weight factors. The collection of detection instances processed through the spatial fusion module is denoted as $\tilde{\mathcal{R}}_{fused}^t$.

*3.4.3 Map Updating Strategy.* To construct spatially continuous map, an updating strategy is needed to merge the upcoming detected instances $\tilde{\mathcal{R}}_{fused}^t = \{(c_i, P_i)\}_{i=1}^{N^t}$ into the maintained global

---

**Algorithm 2** Online Merging algorithm

---

**Input:** global map point sets $\mathcal{P}_G$ and the detected point sets $\mathcal{P}^t$

**Output:** updated global map point sets $\mathcal{P}_G^{\text{update}}$

1: **for** each pair $(\mathcal{P}_G, \mathcal{P}^t, c)$ **do**
2:    **if** $c \in \{\text{boundary}, \text{divider}\}$ **then**
3:       curve_fitting $(\mathcal{P}_G, \mathcal{P}^t)$,
4:    **else if** $c \in \{\text{pedestrian}\}$ **then**
5:       polygon_union $(\mathcal{P}_G, \mathcal{P}^t)$,
6:    **end if**
7: **end for**

---

map $\mathcal{S}_G = \{c_i, P_i\}_{i=1}^{N_G}$. Hence we introduce an online merging algorithm based on Bezier curve fitting [4] applied for instances of class boundary or divider. It determines the point sets order during merging and fits a smoother result. For the pedestrian crossing class, we simply apply a union operation due to their quadrilateral closed shape. Here, we focus on the merging algorithm between a pair of map instance matches of class $c$ and therefore omit the index of it: $(\mathcal{P}_G, \mathcal{P}^t, c)$. Here $\mathcal{P}_G$ denotes the global instance point sets and $\mathcal{P}^t$ denotes the detected instance point sets. This merging process is listed in Algorithm 2 and illustrated by Fig. 5.

## 4 EXPERIMENTS

### 4.1 Experimental Setup

*4.1.1 Dataset.* The nuScenes [5] dataset is one of the most commonly used datasets in the autonomous driving domain. It collects sensor data from approximately 1000 scenes and annotates key samples at a frequency of 2Hz. Each sample provides imagery from six cameras along with the vehicle's global coordinates. Additionally, the dataset includes high-precision map files, allowing for the extraction of map feature data within the current local range for online mapping algorithm development. However, the original partitioning of the nuScenes dataset results in a significant overlap between the training and validation sets, which does not accurately reflect the model's true capability in map feature detection. Therefore, we follow the approach of StreamMapNet [34] to re-partition the nuScenes dataset and perform task evaluations based on this new partitioning.

*4.1.2 Metrics.* Similar to previous related works, we focus on evaluating three types of vector map features: lane dividers, pedestrian crossings, and road boundaries. Given our mapping framework follows a detection-tracking-fusion and update process, we design specific metrics for local detection, instance tracking, and global map construction tasks to validate the advantages of the proposed method in various aspects.

**Detection Metrics.** We conduct evaluations based on two scales of detection range: 100×50 (50m ahead and behind, 25m to each side) and 60×30 (30m ahead and behind, 15m to each side). Average Precision (AP) is used to assess detection capability, with 1.0m, 1.5m, 2.0m as the AP calculation thresholds for large-scale perception tasks, and 0.5m, 1.0m, 1.5m for small-scale tasks.

**Tracking Metrics.** Since the original nuScenes dataset does not provide ID information for map instances, to fit our proposed

instance tracking task, we introduce a post-processing tracking algorithm based on category and distance matching to provide ID labels for training and evaluation. Metrics from the multi-object tracking domain, such as MOTA, MOTP, and IDs, are introduced to evaluate our tracking performance for vector map instances.

**Map Construction Metrics.** A simple merging strategy is employed on annotated data to construct global maps for the evaluation. We then introduce the Chamfer Distance (CD) from HDMapNet [14] to assess the average error in global map construction.

*4.1.3 Baseline Establishment.* As we are the first to evaluate metrics across detection, tracking, and mapping tasks simultaneously, there are not many open-source and similar works available for comparison. This section will briefly introduce the baseline algorithms implemented for the comparison across various metrics.

| Range | Method | $\text{AP}_{ped}$ | $\text{AP}_{div}$ | $\text{AP}_{bound}$ | mAP |
|---|---|---|---|---|---|
| $60 \times 30\,m$ | VectorMapNet [18] | 15.8 | 17.0 | 21.2 | 18.0 |
| | MapTR [17] | 6.4 | 20.7 | 35.5 | 20.9 |
| | StreamMapNet [34] | 29.6 | **30.1** | **41.9** | 33.9 |
| | IC-Mapper(Ours) | **34.4** | 29.3 | 41.4 | **35.0** |
| $100 \times 50\,m$ | VectorMapNet [18] | 12.0 | 8.1 | 6.3 | 8.8 |
| | MapTR [17] | 8.3 | 16.0 | 20.0 | 14.8 |
| | StreamMapNet [34] | 24.8 | 19.6 | 24.7 | 23.0 |
| | IC-Mapper (Ours) | **25.3** | **22.5** | **25.9** | **24.6** |

**Table 1: Detection performance comparison with baseline methods on the new nuScenes split at both** $30\,m$ **and** $50\,m$ **perception ranges. IC-Mapper outperforms existing methods.**

| Range | Method | $\text{CD}_{ped}$ | $\text{CD}_{div}$ | $\text{CD}_{bound}$ | mCD |
|---|---|---|---|---|---|
| $60 \times 30\,m$ | Post-track & VMA-merge [9] | 4.93 | 3.28 | 1.52 | 3.24 |
| | Cluster & Fit [14] | 5.16 | 3.56 | 2.39 | 3.70 |
| | PolyMerge [26] | 4.47 | 3.60 | 1.60 | 3.22 |
| | IC-Mapper (Ours) | **4.27** | **2.87** | **1.23** | **2.79** |
| $100 \times 50\,m$ | Post-track & VMA-merge [9] | 7.89 | 4.46 | 3.38 | 5.24 |
| | Cluster & Fit [14] | 5.64 | 3.82 | 2.80 | 4.09 |
| | PolyMerge [26] | 7.06 | 4.06 | 2.03 | 4.38 |
| | IC-Mapper (Ours) | **5.44** | **3.86** | **1.77** | **3.69** |

**Table 2: Global mapping performance comparison with baseline methods on the new nuScenes split at both** $30\,m$ **and** $50\,m$ **perception ranges. IC-Mapper outperforms existing methods.**

**Detection Baselines.** Three detection baselines are selected for the comparison because of their state-of-the-art performance. VectorMapNet [18] and MapTR [17] are two single-frame-based end-to-end vector map detection networks, whereas StreamMapNet [34] utilizes multi-frame surround-view images as its input.

**Tracking Baselines.** Due to the lack of prior work on visual tracking of vector map features, we reproduce two typical tracking algorithms for comparison. The first, referred to as Post-Track, which involves post-processing matching ns utilizes categories and positional distances from detection results. The second is MOTR-Track, similar to the MOTR [35], which is an end-to-end detection and tracking approach where a subset of the detection queries in the current frame are initialized using tracking queries. This method

| Method | Ped Crossing | | | Divider | | | Boundary | | | mAP |
|--------|------|------|-----------|------|------|-----------|------|------|-----------|------|
|        | MOTA↑ | MOTP↓ | ID-switch↓ | MOTA↑ | MOTP↓ | ID-switch↓ | MOTA↑ | MOTP↓ | ID-switch↓ | |
| Post-Track | 0.54 | **2.72** | 1.72 | **0.50** | **4.19** | 7.75 | **0.80** | 1.83 | 3.41 | 34.1 |
| MOTR-Track | 0.36 | 4.31 | **1.23** | 0.25 | 6.59 | 3.18 | 0.60 | 4.35 | 2.74 | 22.7 |
| IC-Mapper | **0.57** | 3.36 | 1.53 | 0.33 | 4.41 | **2.33** | 0.74 | **1.77** | **1.59** | **35.0** |

**Table 3: Tracking performance comparison with baseline methods on the new nuScenes split at $30\,m$ perception ranges.**

explicitly assigns ground truth instances based on the ID value supervision.

**Mapping Baselines.** Three algorithms have been selected as baselines for the mapping task. The first builds upon a detector trained using StreamMapNet, performing offline tracking and merging features based on the merging strategy in VMA-merge [9]. The second algorithm, inspired by HDMapNet [14], initially clusters point sets detected across an entire image sequence to distinguish instances and subsequently fits the point sets of the same instances. PolyMerge [26] proposes to perform online map updates based on a rule-based algorithm. We reproduce the algorithm and find that our IC-Mapper produces better results under the Chamfer Distance (CD) metric.

*4.1.4 Implementation Details.* Our training approach is conducted in two stages. In the first stage, we jointly train the detection and tracking modules for 24 epochs on 8 GPUs with a batch size of 32, using the AdamW optimizer and set the learning rate to $lr = 5 \times 10^{-4}$. In the second stage, we freeze the parameters of the detection and tracking network modules and train the fusion module at a learning rate of $0.1 \times lr$. Regarding the data selection strategy, in the first stage, we follow the same approach as StreamMapNet, randomly splitting each video sequence into two parts. In the second stage, we sequentially load and train on the entire sequence. We abandoned the single-frame pre-training originally used in StreamMapNet because we found that incorporating the temporal association module allows for greater benefits when training directly on multi-frame data from scratch.

## 4.2 Modular Results

*4.2.1 Detection Results.* We design temporal association and spatial fusion modules for map instance tracking and global map construction tasks. However, we have observed that the IC-Mapper also achieves significant accuracy improvements in detection tasks compared to previous algorithms. As shown in Table 1, our method surpasses existing map instance detection algorithms across various detection range configurations. Compared to StreamMapNet [34], our method shows an average mAP improvement of 1.1 and 1.6 in the detection ranges of 60x30 and 100x50, respectively.

*4.2.2 Tracking Results.* Table 3 demonstrates the performance differences in map feature tracking tasks between our algorithm and other algorithms. Traditional tracking algorithms maintain high levels of MOTA, while another end-to-end detection and tracking algorithm (MOTR) achieves fewer ID Switches.The strengths of our method are reflected in its solid performance across all tracking metrics. Additionally, for multitask learning, the interference between different tasks is an important measure, and our algorithm

enhances the accuracy of the original detection tasks while maintaining high tracking performance, achieving a 12.3 higher mAP in detection compared to MOTR.

*4.2.3 Mapping Results.* Table 2 compares the performance of different map construction methods, and the visualization results on select sequences are shown in Fig. 6. Traditional tracking methods and VMA-based [9] instances substitution strategy struggle to effectively merge multi-frame detection results in the spatial dimension, often leading to localized omissions. Clustering and PolyMerge [26], two mapping algorithms that do not rely on instance tracking, find it challenging to accurately capture the temporal correspondence of map instances, resulting in erroneous connections. Our end-to-end detection-tracking-fusion algorithm models both time and space dimensions using deep learning techniques, achieving superior mapping results. Additionally, our use of Bézier curve-based fitting also contributes to the mapping task metrics, with a detailed analysis available in Fig. 7.

## 4.3 Ablation Study

*4.3.1 Effect of Temporal and Spatial Module.* In Table 4, we conduct ablation experiments to assess the impact of the temporal association module and spatial fusion module on detection accuracy. The experimental results indicate that the temporal association and spatial fusion modules designed for our tracking and mapping modules also significantly enhance the detection metrics. This improvement not only boosts the performance of the current tasks but also offers substantial advantages for subsequent mapping tasks, laying a strong foundation for more accurate and comprehensive map construction.

**Table 4: Detection Accuracy, w.r.t., Temporal and Spatial Modules.**

| Method | $AP_{ped}$ | $AP_{div}$ | $AP_{bound}$ | mAP |
|--------|-----------|-----------|-------------|------|
| Base Detector | 29.62 | 30.18 | 38.98 | 32.92 |
| +temporal module | 33.93 | 29.58 | 41.33 | 34.95 |
| +spacial module | 34.43 | 29.25 | 41.43 | 35.04 |

*4.3.2 Effect of designs in Association Modules.* We conduct evaluations on the geometric and feature-dimensional association matrices of the temporal module. The results show that the model in both dimensions achieves good tracking performance. Additionally, the design based on query features further enhances the detection task metrics. In our approach, the fusion of these two types of metrics is employed, resulting in a strong performance in both detection and tracking indicators.

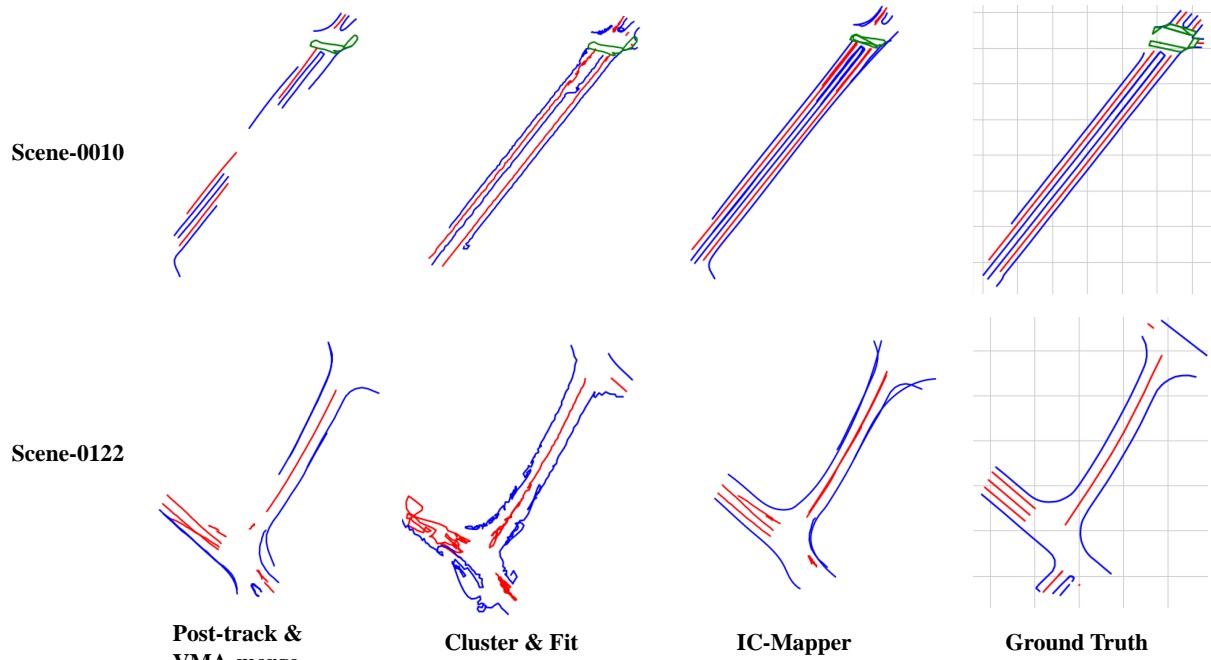

**Figure 6: This visualization presents a comparative analysis of our IC-Mapper against VMA-merge [9] and Cluster & Fit [14] on the scene-0010 and scene-0122 in the dataset. It illustrates that our method excels in capturing finer details of the map, outperforming other algorithms in terms of detail accuracy and completeness.**

**Table 5: The tracking performance concerning different temporal association models, where the "-geo", "-query", and "all" means the calculations based solely on the geometric dimensions, the query feature dimensions, and a fusion of both dimensions for the association, respectively.**

| Method | MOTA | MOTP | ID-switch | mAP |
|---|---|---|---|---|
| Ours-geo | 0.57 | 3.12 | 1.83 | 34.65 |
| Ours-query | 0.57 | 3.23 | 1.91 | 35.17 |
| Ours-all | 0.54 | 3.17 | 1.81 | 35.04 |

*4.3.3 Smoothing Parameter in Curve Fitting.* As shown in Fig. 7, when fitting polyline instances using Bézier curves, the correct choice of the smoothing parameter $s$ can balance accuracy and smoothness. Both overly low and overly high smoothing parameters can result in a loss of curve fitting accuracy. Therefore, we recommend using a smoothing parameter $s$ within [0.1,1].

## 5 CONCLUSION

We propose an instance-centric end-to-end detection-tracking-fusion framework for vision-based online construction of vectorized maps. We designed a temporal association module to match instances across frames and introduced a spatial fusion module to merge previous maps with current detection results. Based on such temporal and spatial modeling, our method achieves leading levels in metrics across detection, tracking, and mapping. Furthermore, we also

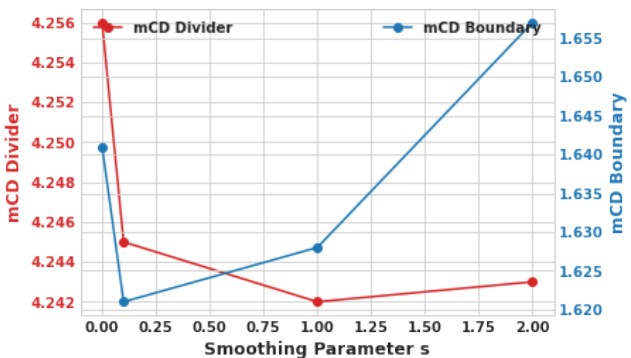

**Figure 7: This graph illustrates the variation in fitting accuracy for lane dividers and road boundaries with a smoothing function of Bézier curves with the parameter $s$. The x-axis represents the smoothing parameter $s$, ranging from 0 to 2.0, while the y-axis measures the accuracy of the curve fitting.**

encourage the community to ponder and explore more vectorized mapping tasks in long temporal sequences and large spatial scales.

**Limitations:** We have introduced an end-to-end framework for detection, tracking, and map construction for the first time. However, the modeling of relationships between different sub-modules requires further exploration to enhance overall performance and integration. In addition, more datasets (besides the field-established and highly-quality annotated nuScenes) are preferred for the enhanced establishment of the method performance.

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

Received 20 February 2007; revised 12 March 2009; accepted 5 June 2009

