# OpenReview forum: "IC-Mapper: Instance-Centric Spatio-Temporal Modeling for Online Vectorized Map Construction"
_acmmm.org/ACMMM/2024/Conference — MM2024 Poster_

### Official Review · Reviewer_VK4h · 2024-05-22

**Rating:** 3
**Confidence:** 3

**Summary:**

This paper proposes an instance-centric detection-tracking-fusion framework for global map construction via sequence temporal input images. Based on basic detection, temporal association module is utilized to obtain the matching correspondence for instance tracking, while instance spatial fusion module is proposed to achieve map update and expansion. On nuScenes dataset, experiments on detection, tracking and global map construction demonstrate the superior performance compared with previous sotas.

**Strengths:**

1. This paper is well-written and easy-to-follow.
2. Detection-tracking-fusion strategy seems to be interesting and reasonable for global map construction.
3. This paper achieves online mapping updating and construction, which can be practical for industries like mapping and autonomous driving companies

**Limitations:**

1. For comprehensive assessment, experiments on Argoverse dataset are required. Besides, what is the inference speed and memory usage for each part of your strategy?
2. Tracking and fusion are helpful strategy for utilizing inter-frame information, but in table 1, compare with previous sotas, the improvement seems limited and not as expected.
3. The topK query selection from previous frame and fusion with the current strategy is similar to the instance temporal fusion and has already been used in StreamMapNet [1].
4. Bezier curve fitting may lose efficacy, which is hard to fit for some sudden change road structures or sharp corners. How do you handle such conditions? More visualizations for such scenes are required to show the global result.

[1] Tianyuan Yuan, Yicheng Liu, Yue Wang, Yilun Wang, and Hang Zhao. 2024. Streammapnet: Streaming mapping network for vectorized online hd map construction. In WACV. 7356–7365

**Suitability:**

2

---

### Official Review · Reviewer_uvf1 · 2024-05-25

**Rating:** 3
**Confidence:** 3

**Summary:**

The authors propose IC-Mapper, an instance-centric online mapping framework, which comprises two primary components: 1) Instance-centric temporal association module. The former is designed for measuring the detection queries of adjacent frames in both feature and geometric dimensions to obtain the matching correspondence between instances across frames. The latter is responsible for performing point sampling on the historical global map from a spatial dimension and integrating it with the detection results of instances corresponding to the current frame to achieve real-time expansion and update of the map.

**Strengths:**

1. The authors have implemented an end-to-end detection-tracking-fusion process that enables the construction of global maps, incorporating temporal tracking and spatial fusion modules.
2. The authors are among the pioneers to incorporate all online mapping tasks in one network including detection, tracking, and the global map update.

**Limitations:**

1. It is suggested that the authors discuss any instances of lower performance relative to existing methods as observed in Table 1 and Table 3, offering a thorough analysis and potential explanations for these results.
2. It would be beneficial for the authors to elucidate the factors contributing to the performance decline of AP_{div} following the integration of the temporal and spatial modules, as detailed in the ablation study of Table 4, accompanied by a comprehensive discussion and analysis.
3. Table 4 indicates a comparatively marginal enhancement in performance attributed to the spatial module when juxtaposed with the temporal module. An in-depth discussion and analysis from the authors regarding this disparity would be insightful.
4. I would raise my score if the authors could effectively address my concerns.

**Suitability:**

3

---

### Official Review · Reviewer_BpCy · 2024-05-27

**Rating:** 3
**Confidence:** 4

**Summary:**

This paper presents IC-Mapper, a new framework for global temporal consistent vector map detection. It can detect vector maps, and merge them into a global map. First, IC-Mapper detects vector maps of each frame following StreamMapNet, which can aggregate information of multiple frames for local map detection. Then, it proposes a Instance-CentricTemporalAssociation module for map instance association. Finally, associated instances are merged based on the proposed spatial fusion module. Experiments on the large NuScenes dataset are conducted, and the effectiveness of the model is proved.

**Strengths:**

This paper is clear to read and easy to follow. The methodology is clearly claimed and multiple figures are provided for readers. The general design is solid and the motivation is convincing.

**Limitations:**

This paper is generally fine for me, but there are still some concerns about the experiments.

1. Is there any filtering steps in the section 3.4 (spatial fusion)? I believe the vector map detection of each step is not temporal consistent, so what if the associated instances have very different shapes (Or directly incorrect association)? Directly merging them is not a very smart method for me. Have you considered adding the confidence score into the merge algorithm for better performance?

2. IC-Mapper is based on SpatialMapNet. In Figure 1, IC-Mapper obtains a slightly higher score than SpatialMapNet. Are there any modifications to the SptialMapNet network that can explain the improvement?

3. It seems that Post-track has a very similar performance with IC-Mapper (34.1, 35.0), why does Post-track & VMA-merge have that bad results in Table 2? Is it because of the proposed IC-Mapper spatial fusion module? What if you use Post-track in IC-Mapper? In Table 4, it seems that the spatial module does not make the main contribution to improvement (+0.09mAP). I am confused here. Could you please do more experiments about different associations and merging method combinations? E.g., IC-Mapper association + VMA-merge; Post-track + iC-Mapper merge.

4. Following question 3, how could IC-Mapper merge outperform VMA-merge that far? Considering that the IC-Mapper merge is quite straightforward.

5. Do all baselines have exactly the same efforts for optimization? In Figure 6, if Post-track has quite good association performance, how could VMA-merge have such bad merging results? Besides, I noticed IC-Mapper has many duplicated detections (detecting one line two times) and baseline methods do not have such an issue. Could you please do some discussion here?

6. What about the model efficiency? Can it run online?

7. Cite more recent papers will make your paper more concrete and convincing, even if you do not compare them in your paper. Like
[1] Liao, Bencheng, et al. "Maptrv2: An end-to-end framework for online vectorized hd map construction." arXiv preprint arXiv:2308.05736 (2023).
[2] Xu, Zhenhua, Kenneth KY Wong, and Hengshuang Zhao. "Insightmapper: A closer look at inner-instance information for vectorized high-definition mapping." arXiv preprint arXiv:2308.08543 (2023).
[3] Ding, Wenjie, et al. "Pivotnet: Vectorized pivot learning for end-to-end hd map construction." Proceedings of the IEEE/CVF International Conference on Computer Vision. 2023.
[4] Xie, Ziyang, Ziqi Pang, and Yu-Xiong Wang. "Mv-map: Offboard hd-map generation with multi-view consistency." Proceedings of the IEEE/CVF International Conference on Computer Vision. 2023.

In conclusion, I mainly have some concerns about the experiments. The authors may want to provide more information about it. I will raise my score if some of the aforementioned concerns are solved.

**Suitability:**

3

---

### Meta-Review · Area_Chair_KeGQ · 2024-07-03

**Recommendation:** Accept (Poster)
**Confidence:** 2

**Metareview:**

The proposal concerns an instance-centric detection-tracking-fusion framework for global map construction based on sequence temporal input images. According to the 3 reviewers, the article is well-written and clear, offering contributions, but with weaknesses mainly at the level of experimentation, which led them to a decision around borderline/weak accept or reject. It seems to me that the answers provided in the rebuttal clarify the reviewers' expectations (somewhat surprisingly, this is not what R2 and R3 say), and that the paper could therefore be accepted, probably as a poster, given that the contributions highlighted are not major, but remain relevant to the ACM MM community.